# Exploring the Computational Effects of Advanced Deep Neural Networks on Logical and Activity Learning for Enhanced Thinking Skills

**Deming Li** [1,*] , **Kellyt D. Ortegas** [2] **and Marvin White** [3]

1    School of Education, Jilin International Studies University, Changchun 130117, China
2    Merced College, Merced, CA 95348, USA
3    Department of Information Engineering, Southern University and A&M College,
     Baton Rouge, LA 70813, USA
*    Correspondence: lideming@jisu.edu.cn

**Abstract:** The Logical and Activity Learning for Enhanced Thinking Skills (LAL) method is an educational approach that fosters the development of critical thinking, problem-solving, and decision-making abilities in students using practical, experiential learning activities. Although LAL has demonstrated favorable effects on children's cognitive growth, it presents various obstacles, including the requirement for tailored instruction and the complexity of tracking advancement. The present study presents a model known as the Deep Neural Networks-based Logical and Activity Learning Model (DNN-LALM) as a potential solution to tackle the challenges above. The DNN-LALM employs sophisticated machine learning methodologies to offer tailored instruction and assessment tracking, and enhanced proficiency in cognitive and task-oriented activities. The model under consideration has been assessed using a dataset comprising cognitive assessments of children. The findings indicate noteworthy enhancements in accuracy, precision, and recall. The model above attained a 93% accuracy rate in detecting logical patterns and an 87% precision rate in forecasting activity outcomes. The findings of this study indicate that the implementation of DNN-LALM can augment the efficacy of LAL in fostering cognitive growth, thereby facilitating improved monitoring of children's advancement by educators and parents. The model under consideration can transform the approach toward LAL in educational environments, facilitating more individualized and efficacious learning opportunities for children.

**Keywords:** logical and activity learning; enhanced thinking skills; computational effects; deep neural networks





## 1. Introduction to Logical and Activity Learning

Using logical and activity-based learning techniques to augment cognitive abilities is a pioneering educational methodology that prioritizes cultivating critical thinking, problem-solving, and decision-making proficiencies in young learners [1,2]. The objective is to equip pupils with the necessary skills to become proficient and accountable community constituents, capable of addressing intricate predicaments through rational thinking and ingenuity. The methodology prioritizes experiential education, whereby youngsters participate in practical tasks and drills replicating authentic situations, facilitating learning through experimentation and engagement [3,4].

The salient characteristics of logical and activity-based learning encompass the utilization of interactive multimedia, visual aids, and cooperative learning settings [5]. Through integrating these components, learners can cultivate a more profound comprehension of intricate principles and notions and employ them in practical scenarios. Furthermore, this methodology advocates for a pedagogical approach that emphasizes active engagement,

whereby children are motivated to inquire, investigate, and test hypotheses, cultivating their inquisitiveness and ingenuity.

In contemporary times, there is a growing urgency for incorporating logical and activity-based learning methods, owing to the fast-paced and dynamic nature of the world [6,7]. In light of intensifying competition in the job market and the rapid progress of technology, the capacity to engage in critical and creative thinking is increasingly indispensable. The methodology, as mentioned earlier, facilitates the cultivation of a growth-oriented mentality among pupils, empowering them to confront obstacles and derive knowledge from their errors, both of which are fundamental attributes in any domain.

The implementation of this approach is accompanied by various challenges, such as the requirement for proficient educators capable of effectively facilitating the learning process, insufficient resources and infrastructure, and the absence of established frameworks and evaluation techniques to gauge learning outcomes. Advanced technological solutions, such as Deep Neural Networks (DNNs) [8], can improve the efficacy of logical and activity learning, thereby addressing the challenges above.

DNNs are machine learning algorithms that draw inspiration from the human brain's structure and function. The entities in question consist of numerous strata of interlinked nodes that process and convert input data. DNNs can acquire intricate data representations applicable in diverse visual perception, linguistic analysis, and auditory comprehension domains. In a research-based learning strategy, students seek for and make use of a variety of resources, materials, and texts to investigate issues that are meaningful to them. By reading and learning new words, students improve their ability to discover, analyze, organize, and evaluate information and ideas.

Within the Logical and Activity Learning (LAL) framework aimed at improving cognitive abilities, DNNs can be utilized to develop sophisticated systems capable of tailoring and customizing educational experiences to meet the unique needs of individual learners [9,10]. DNNs can analyze student performance data and generate tailored recommendations for enhancing academic outcomes. Interactive and engaging learning experiences can be facilitated through gamification and virtual environments.

The capacity of DNNs in LAL to acquire knowledge from vast datasets is a significant attribute. DNNs of many different kinds enable models to develop greater efficiency at learning complicated features and executing more intense computational tasks, and to conduct increasingly complex operations concurrently. Developing a career based on DNN-LALM requires the application of critical thinking abilities, like the ability to weigh the advantages and disadvantages, identify root causes, and develop up with original solutions.

Despite the potential advantages of employing Logical and Activity Learning for Enhanced Thinking Skills, several obstacles require attention, including the absence of tailored learning, restricted educator resources, and the need for scalable and adaptable learning frameworks. The challenges of personalized and scalable learning, optimization of learning outcomes through data-driven approaches, and adaptability to individual student needs can be addressed by the Deep Neural Networks-based Logical and Activity Learning Model (DNN-LALM). The implementation of DNN-LALM has the potential to address the obstacles related to LAL, leading to improved cognitive abilities and a more productive educational setting for learners. The limitation of DNN overfitting is a situation where a machine learning model performs badly on fresh, ambiguous data because it has to be simplified and was trained too effectively on the training data.

The primary contributions are listed below:

- The initial stage entails the development of DNN architectures featuring diverse layer configurations to enhance cognitive and behavioral learning.
- The second stage involves the assessment of student performance through the utilization of DNN models trained and tested on the PISA dataset, followed by analysis utilizing the EDM Toolbox.

- The present study used the Educational Data Mining (EDM) Toolbox and the Programme for International Student Assessment (PISA) dataset to evaluate student performance by implementing DNNs.
- The DNN-LALM that can help students become more attentive and focused in an active learning environment, have more meaningful learning experiences, achieve greater levels of performance, and become more motivated to exercise higher-level critical thinking abilities because of such a setting.

The following sections are organized in the next section: Section 2 presents a comprehensive overview of the relevant literature and background information about applying DNNs in educational settings to improve learning outcomes. Section 3 proposes a DNN-based Logical and Activity Learning Model to analyze student performance utilizing the Programme for International Student Assessment dataset. Section 4 presents the simulation analysis and outcomes of the DNN-LALM. The framework is trained and tested on the PISA dataset. Section 5 summarizes the research findings and offers potential avenues for further improving the proposed DNN-LALM.

## 2. Background and Literature Survey

The literature review for Logical and Activity Learning delves into the extant scholarship about the instruction of cognitive abilities via interactive exercises and logical deduction. This paper analyzes the obstacles conventional educational approaches encounter and suggests using sophisticated technologies, such as Deep Neural Networks, to augment the learning process.

Lin et al. introduced a novel smart toy system that utilizes game-based approaches to augment the computational thinking skills of young children in preschool [11]. The methodology involved a gamified framework employing intelligent playthings to impart fundamental computational principles to juveniles. The research findings indicated favorable outcomes regarding children's academic achievements and involvement.

Nurbekova et al. introduced a pedagogical strategy centered on project-based learning to instruct students in developing mobile applications utilizing visualization technology [12]. The objective of the research was to improve learners' abilities in the domain of mobile application development and critical thinking through the utilization of a project-oriented methodology. The suggested method yielded a noteworthy enhancement in academic achievements and aptitude for resolving students' problems.

Wati et al. has suggested an approach to enhance the mathematical-logic learning ability of young children through game-based learning [13]. The research entailed the creation of an instructional game dubbed "LOP Game," which centered on mathematical logic principles. The findings indicate that game-based methodology significantly enhanced children's aptitude for learning mathematical logic.

Çiftci et al. examined the impact of coding subjects on the cognitive aptitudes and problem-solving proficiencies of young students in preschool [14]. The research involved the introduction of a coding curriculum in a preschool environment, which yielded favorable outcomes in the form of enhanced cognitive capabilities and improved problem-solving proficiencies among the children.

Aminov et al. surveyed the challenges associated with creating instructional resources that facilitate student engagement in the learning process within the context of education [15]. The research examined the significance of employing contemporary technologies and methodologies to enhance students' academic achievements and involvement. This research fails due to the need for examples and evidence to support the findings.

The present study suggests interactive approaches for instructing Russian literature in educational institutions where Uzbek language acquisition occurs [16]. The necessity of employing this approach stems from the difficulties encountered by students of the Uzbek language in comprehending and valuing Russian literary works. The proposed methodology uses interactive pedagogical techniques such as dramatization, role-playing, and discussions to involve students in active learning and interpreting Russian literature actively.

This paper examines the utilization of blended learning to enhance university-level students' critical thinking and communication abilities [17]. The study's findings indicate that integrating face-to-face and online learning, commonly called blended learning, is viable for fostering students' critical thinking and communication competencies. Blended learning is characterized by its flexible nature, personalized approach to education, and provision of digital resources.

The study suggests the implementation of Project-Based Learning-Literacy (PBL-L) as a means to enhance the mathematical reasoning skills of elementary school students [18]. The approach entails involving learners in authentic problem-solving tasks that necessitate the utilization of mathematical principles and proficiencies. The simulation findings indicate that implementing PBL-L significantly positively impacts enhancing students' mathematical reasoning skills.

This study examines the inquiry-based learning approach's efficacy in enhancing pre-service teachers' metacognitive knowledge and awareness [19]. The research findings indicate that the inquiry-based learning approach, characterized by self-directed and active learning, fosters metacognitive knowledge and understanding among aspiring educators. Inquiry-based learning is characterized by the incorporation of experiential activities, cooperative learning, and analytical thinking to facilitate problem-solving.

The present study suggests creating educational tools utilizing the principles of Realistic Mathematics Education (RME) to enhance students' spatial aptitude and motivation [20]. The proposed methodology entails the utilization of practical problem-solving scenarios, tangible objects, and graphical illustrations as pedagogical tools for imparting mathematical concepts. The simulation findings indicate that RME-based learning devices significantly enhance students' spatial ability and motivation.

The literature review investigated diverse approaches to augmenting cognitive abilities in children, such as utilizing smart toys based on games, adopting project-based learning, and implementing inquiry-based learning models. Nevertheless, the obstacles encountered in the execution of these techniques underscore the necessity for a more sophisticated methodology, such as the DNN-LALM that has been suggested. The study's findings indicate that DNN-LALM possesses characteristics that enable it to effectively tackle the obstacles above and improve cognitive and motor skill acquisition in young individuals.

## 3. Proposed Deep Neural Networks-Based Logical and Activity Learning Model

The objective of the proposed approach is to investigate the influence of sophisticated deep neural networks on the acquisition of logical and activity-based knowledge to improve cognitive abilities. The methodology comprises a dual-phase strategy, wherein the initial phase entails developing DNN models by utilizing diverse datasets. The second step involves the assessment of the efficacy of the trained models in the context of logical and activity-based learning tasks. The outcomes of the investigation suggest that DNN-LALM can increase the efficacy of LAL in promoting children's cognitive development, allowing for better tracking of children's progress by teachers and parents. The proposed methodology has the potential to drastically alter how LAL is seen in classrooms, giving students more chances for meaningful, targeted instruction. However, instead of sitting passive during teacher talks, children participating in activity-based learning are encouraged to take an active role in their own education by carrying out planned activities. The main difference between teaching strategies and teaching techniques is that the former center on the way knowledge is delivered to students, and the latter emphasize how instructors may best achieve their learning objectives.

With its structured design, the proposed system is displayed in Figure 1.

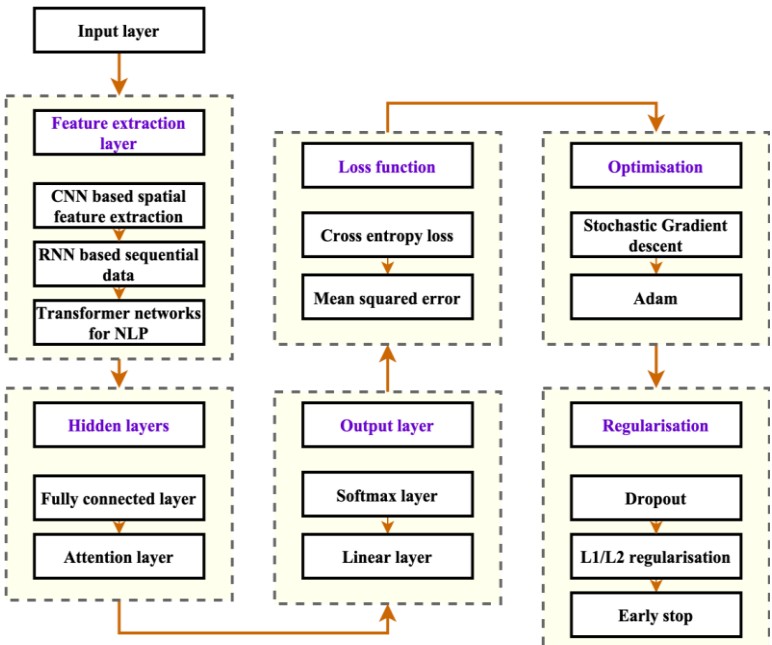

**Figure 1.** The proposed DNN-LALM design.

*3.1. Input Layer*

The initial layer of a neural network is responsible for receiving the input data from LAL, commonly represented as a tensor with many dimensions. The input layer lacks any trainable parameters, and its primary role is to transmit the input LAL data from the student to the second layer.

The initial layer of the system functions to receive LAL input data and transmit it to the second layer without any alterations. The input data is commonly expressed as a tensor with high dimensions, wherein each size pertains to distinct features or constituents of the input.

The present architecture is designed to accommodate diverse types of LAL data in the input layer, including but not limited to sensor readings, textual input, and image data. The input tensor possesses the image data's configuration (height, width, channels). Height and width denote the image's dimensions, while channels signify the number of color channels present.

The input layer lacks any trainable variables for LAL; its primary role is transmitting the input to the second layer. The result of the layer is equivalent to the input data, which is expressed as a tensor. The input layer can be theoretically reflected in Equation (1).

$$X = \left[ x_{i_1}, x_{i_2}, \cdots, x_{i_n} \right]_{i_1=1, i_2=1, \cdots, i_n=1}^{i_1=h, i_2=w, \cdots, i_n=c} \tag{1}$$

Consider a tensor $X$ with dimensions $(h, w, c)$, where $x_{i_1}, x_{i_2}, \cdots, x_{i_n}$ denotes the value of the input at location $i_1, i_2, \cdots, i_n$. Preprocessing or normalization of input data is a common practice before its utilization in a neural network. In image data, it is possible to rescale the pixel values to a range of [0, 1] or standardize them to achieve zero mean and unit variance. The implementation of a preprocessing step has the potential to enhance the network's efficiency and convergence.

*3.2. Feature Extraction Layer*

The layer in question extracts pertinent features from the LAL input data. The composition of sub-blocks within this layer is contingent upon the nature of the input data. However, it typically encompasses Convolutional Neural Networks (CNNs), Recurrent Neural Networks (RNNs), and Transformer Networks.

### 3.2.1. CNN

CNNs are neural networks frequently employed to process image data. Convolutional filters are utilized to extract spatial characteristics from the input image. The typical outcome of a CNN layer is a feature map, wherein each LAL element of the map signifies the activation of a specific filtering to a particular point within the input picture.

The process of convolution can be scientifically reflected using Equation (2).

$$y_{i,j} = \sum_{k=0}^{m-1} \sum_{l=0}^{n-1} x_{i+k,j+l} \times w_{k,l} \tag{2}$$

The input tensor, denoted as $x$, is convolved with a filter, represented by $w$, resulting in an output tensor, denoted as $y$. The summation operation is performed over the size of the filter. The filter is generally acquired through the training phase for LAL and can be conceptualized as a collection of coefficients employed to the LAL input tensor.

### 3.2.2. RNN

RNNs are neural networks frequently employed for processing sequential data. A sequence of hidden states is used to capture temporal dependencies present in the input data. The hidden state is the typical output of an RNN layer at each time step, and it is commonly utilized as input for the time step within the network.

Calculating an RNN hidden state at a given time step $t$ can be expressed mathematically in Equation (3).

$$h_t = \sigma\{W_{xh}x_t + W_{hh}h_{t-1} + b_h\} \tag{3}$$

The equation above represents a mathematical model for a recurrent neural network, where $x_t$ denotes the input at a given time step $t$, $h_{t-1}$ represents the previous hidden state, $W_{xh}$ and $W_{hh}$. are weight matrices, $b_h$ is a bias vector, and $\sigma$ shows an activating feature.

### 3.2.3. Transformer Networks

A Transformer Network is an overall neural network architecture utilized in various natural language processing applications. Self-attention mechanisms capture the interdependencies among distinct tokens present in the input text. The typical outcome of a Transformer layer is a series of concealed states, wherein each state denotes a specific token within the input text.

The computation of a Transformer layer can be scientifically reflected using Equation (4).

$$A(Q, K, V) = softmax\left(\frac{QK^T}{\sqrt{d_k}}\right)V \tag{4}$$

$Q$, $K$, and $V$ are related to the queries, keys, and vectors. The dimensionality of the vital vector is represented by $d_k$. The attention mechanism calculates a summation of the vectors of value, which is weighted by the similarity between the vectors of query and key.

The sub-blocks employ diverse mathematical operations and methodologies to capture spatial and temporal interdependencies in the input data and produce more advanced data representations.

### 3.3. Hidden Layers

The Hidden Layers are responsible for intricate calculations on the extracted LAL features to acquire more advanced representations of the LAL input data. The layer contains sub-blocks, which have the potential to encompass.

### 3.3.1. Fully Connected Layers

Fully Connected Layers establish connections between each neuron in the layer and every neuron in the next layer. This facilitates the acquisition of intricate nonlinear associations between the LAL input and output by the network. The computation of the production of a Fully Connected Layer can be expressed in Equation (5).

$$y = \sigma(Wx + b) \tag{5}$$

The equation above involves the weight matrix denoted by $W$, the input vector represented by $x$, the bias vector indicated by $b$, and an activation function characterized by $\sigma$. The matrix denoted by $W$ possesses a shape of $(n_{out}, n_{in})$, where $n_{in}$ represents the count of neurons in the preceding layer, and $n_{out}$ denotes the count of neurons. The vector $b$, which represents bias, possesses a shape of $n_{out}$.

The result yielded by a Fully Connected Layer is a vector with a magnitude of $n_{out}$ wherein each constituent represents the output of a neuron present in the current layer for LAL.

### 3.3.2. Attention Layers

The Attention Layers employ attention strategies to concentrate on pertinent segments of the input data, enabling the network to acquire the ability to focus on significant characteristics. The computation of the output of an Attention Layer can be expressed in Equation (6).

$$y = \sum_{i=1}^{n_{in}} \propto_i x_i \tag{6}$$

The equation involves the i-th feature vector denoted as $x_i$ in the input, where $n_{in}$ represents the total number of features present in the information. The attention weight assigned to the i-th feature vector is defined by the symbol $\propto_i$. The computation of attention weights involves utilizing a softmax function, expressed in Equation (7).

$$\propto_i = \frac{\exp(e_i)}{\sum_{j=1}^{n_{in}} \exp(e_i)} \tag{7}$$

The scalar energy value, denoted as $e_i$, is linked to the i-th feature vector. Diverse techniques exist for computing energy values, including dot product attention, additive attention, and multi-head attention.

### 3.4. Output Layer

The Output Layer is accountable for generating the LAL output of the network. The layer contains sub-blocks, which have the potential to encompass.

### 3.4.1. Softmax Layer

The Softmax Layer is a crucial component in categorization tasks as it generates a probability distribution across the various classes of LAL. The computation of the output of a Softmax Layer can be expressed in Equation (8).

$$y_i = \frac{\exp(z_i)}{\sum_{j=1}^{n_{out}} \exp(z_i)} \tag{8}$$

The i-th component of the input vector is represented by $z_i$, while $n_{out}$ denotes the total number of classes. The softmax function guarantees that the resultant probabilities for LAL of the output sum to 1.

### 3.4.2. Linear Layer

The Linear Layer generates continuous results in regression tasks for LAL. The computation of the result of a Linear Layer can be expressed in Equation (9).

$$y = Wx + b \tag{9}$$

The weight matrix is denoted by $W$, the input vector is represented by $x$, and the bias vector is indicated by $b$. The resultant variable $y$ is a vector that exhibits continuity.

### 3.5. Loss Function

Loss Function measures the difference between the predicted and actual outcomes in a machine learning model. The present code block calculates the discrepancy between the anticipated and actual outputs. The layer contains sub-blocks that are capable of inclusion.

### 3.5.1. Cross-Entropy Loss

The sub-block is frequently employed in classification scenarios and quantifies the dissimilarity between the anticipated and actual probability distributions. The term can be defined using Equation (10).

$$L_{CE} = -\frac{1}{N} \sum_{i=1}^{N} \sum_{j=1}^{C} y_{ij} \log(\hat{y}_{ij}) \tag{10}$$

$N$ represents the total number of the specimen, $C$ denotes the total number of classes, $y_{ij}$ indicates the actual probability of the i-th selection being a part of the jth class, and $\hat{y}_{ij}$ represents the anticipated probability of the i-th sample being a part of the jth category.

### 3.5.2. Mean Squared Error (MSE)

The MSE Loss is a frequently employed sub-block in regression tasks, which quantifies the dissimilarity between the predicted and actual outputs. The term can be defined in Equation (11).

$$L_{MSE} = \frac{1}{N} \sum_{i=1}^{N} (y_i - \hat{y}_i)^2 \tag{11}$$

In the given equation, $N$ represents the total number of samples. The variable $y_i$ denotes the actual output of the i-th example, while $\hat{y}_i$ represents the predicted output of the same model.

### 3.6. Optimization Algorithm

The Optimization Algorithm is responsible for adjusting the network weights by the error calculated by the loss function. The layer in question contains sub-blocks, which have the potential to encompass.

### 3.6.1. Stochastic Gradient Descent (SGD)

SGD is a commonly used optimization method, an iterative process that aims to minimize a given objective function by updating the model parameters in the direction of the negative gradient of the process. The method randomly selects a subset of the training samples, mini-batches, to calculate the gradients and upgradation of the variables. This process is repeated until convergence or a stopping criterion is met. SGD is known for its efficiency and scalability, making it a popular choice for large-scale machine learning problems. The weights are updated in this sub-block through the computation of the gradient of the loss function about the coefficients, followed by a movement in the opposite direction.

The term can be defined in Equation (12).

$$\theta_{t+1} = \theta_t \nabla_\theta L(\theta_t) \tag{12}$$

The equation pertains to the weights at a given time, denoted by $\theta_t$, where $\nabla_\theta$ represents the learning rate, and $L(\theta_t)$ signifies the gradient of the loss function about the coefficients at the same time $t$.

3.6.2. Adam

The sub-block pertains to an optimization method for adaptive learning rates designed to calculate different learning rates for various parameters. The term can be defined using Equations (13)–(16).

$$m_{t+1} = \beta_1 m_t + (1 - \beta_1) \nabla_\theta L(\theta_t) \tag{13}$$

$$v_{t+1} = \beta_2 v_t + (1 - \beta_2)(\nabla_\theta L(\theta_t))^2 \tag{14}$$

$$\hat{m}_{t+1} = \frac{m_{t+1}}{1 - \beta_2^{t+1}} \tag{15}$$

$$\theta_{t+1} = \theta_t - \delta \frac{\hat{m}_{t+1}}{\sqrt{\hat{v}_{t+1} + \sigma}} \tag{16}$$

The equation pertains to the weights at time $t$, denoted by $\theta_t$. It involves several variables such as the learning rate $L$, deteriorate rates $\beta_1$ and $\beta_2$ for the first and second instances, first and second instances of gradient denoted by $m_t$ and $v_t$, bias-corrected estimates of the first and second instances characterized by $\hat{m}_{t+1}$ $and$ $v_{t+1}$, learning pace $\delta$, and a small value $\sigma$ that is utilized to prevent dividing by zero.

*3.7. Regularization*

Regularization techniques mitigate overfitting and enhance the network's generalization capacity. The layer comprises sub-blocks, which have the potential to encompass.

3.7.1. Dropout

To prevent over-reliance on specific features, a technique known as a dropout is employed, whereby specific neurons are randomly dropped out during the training process. Its mathematical expression can be formulated in Equations (17) and (18).

$$h_i = f(w_{ij} x_{ij} + b_i) \tag{17}$$

$$h'_i = \begin{cases} h_i & with\ prob\ p \\ 0 & with\ prob\ 1 - p \end{cases} \tag{18}$$

The equation denotes the output of a neuron, represented by the variable $h$, which is determined by the weights assigned to it, marked by $w$, and the bias, denoted by $b$. The activation function, represented by the variable $f$, is applied to the input variable $x$. The probability of retaining a neuron is represented by the variable $p$.

3.7.2. L1/L2 Regularization

The $L1$/ $L2$ regularization technique involves incorporating a penalty term into the loss function, which promotes the minimization of weight values and mitigates the risk of overfitting. The concept can be articulated in Equations (19) and (20).

$$L1 = L(\theta) = Loss\ (\theta) + \delta \sum_{i=1}^{n} |\theta_i| \tag{19}$$

$$L2 = L(\theta) = Loss\ (\theta) + \frac{\delta}{2} \sum_{i=1}^{n} (\theta_i)^2 \tag{20}$$

The equation pertains to the loss function denoted by $L(\theta)$, the weight vector represented by theta, the number of weights indicated by $n$, the loss function is denoted *Loss* $(\theta)$, and the regularization hyperparameter denoted by $\delta$.

### 3.7.3. Early Stopping

Early Stopping involves monitoring the evaluation process during training and stopping the training procedure before it reaches the point of overfitting. This is achieved by setting a threshold for the performance metric, such as validation loss, and stopping the training process when the metric no longer improves beyond the threshold. Early stopping is a widely used technique in machine learning and has been shown to improve the generalization performance of models. The training process is terminated prematurely by this sub-block in the event of an increase in validation loss, thereby preventing overfitting. The concept is articulated using Equation (21).

$$ES = \arg(\min(L_{val}(\theta))) \tag{21}$$

The validation loss is denoted as $L_{val}$, and the weight vector is represented by $\theta$. The cessation of the training process occurs once the validation loss initiates an upward trend.

The architecture under consideration comprises multiple blocks and sub-blocks, encompassing the input layer, feature extraction layer, hidden layers, output layer, loss function block, and regularization block. The initial layer performs pre-processing methodologies to convert unprocessed input data into an appropriate structure for the following layers. Extracting pertinent features from the input data is accomplished by utilizing convolutional and pooling sub-blocks within the feature extraction layer. The concealed strata employ fully connected and attention subunits to acquire more advanced representations of the input information. The output layer generates the ultimate result by utilizing softmax and linear sub-blocks for classification and regression tasks. The loss function module comprises sub-modules, namely cross-entropy loss and MSE loss, which calculate the discrepancy between the predicted and actual outputs. The regularization module employs various sub-modules, including dropout, L1/L2 regularization, and early stopping, to mitigate overfitting and enhance the generalization capacity of the model.

### 3.8. Deep Neural Network Evaluation Method
### 3.8.1. Deep Neural Networks

DNNs are constructed using artificial neural network architecture, characterized by multiple hidden layers and a higher number of network nodes. This distinguishes DNNs from conventional neural networks. The utilization of hidden layers in DNNs facilitates the identification of intrinsic characteristics of the data, leading to an enhancement in the modeling capacity to acquire multiple layers. DNNs can extract shared fundamental characteristics of a given dataset using limited training data and exhibit strong modeling abilities for intricate tasks through the involvement of multiple neurons. The procedure for DNNs is as follows.

Upon completion of data preprocessing, the initialization data is transmitted from the input layer to the initial hidden layer. The functional mapping between the initial hidden layer's input and output is referred to using Equation (22).

$$r_i = fn(w_i x + b_i) \tag{22}$$

Assuming all output values in $r_i$ are derived from the source vector $x$ by applying the activating method $fn$. the weight is denoted $w_i$, and the bias is denoted $b_i$. The output variable is denoted in Equation (23).

$$r_{i,m} = fn\left(\sum_{k=0}^{n-1} w_{i,m,k} x_i + b_{i,m}\right) \tag{23}$$

The weight is denoted $w_{i,m,k}$, the input is denoted $x_i$, and the biasing is denoted $b_{i,m}$. The DNN model's qth concealed layer results, denoted as $r_q$, can be acquired based on the principle of DNis characterized in Equation (24).

$$r_q = fn\left(w_q r_{q-1} + b_q\right) \tag{24}$$

The weight, output variable, and biasing are denoted $w_q$, $r_{q-1}$ and $b_q$. The worth of each constituent within the resultant $r_q$ of the undisclosed stratum of $r_{q,m}$ stratum $Q$ is being referred to in Equation (25).

$$r_{q,m} = fn\left(\sum_{k=0}^{n-1} w_{q,m,k} r_{q-1,i} + b_{q,m}\right) \tag{25}$$

The weight, output variable, and biasing are denoted $w_{q,m,k}$, $r_{q-1,i}$ and $b_{q,m}$. After processing, the input vector $X$ will be conveyed to the output layer. The outcome is presented in Equation (26).

$$y = c(w_{n+1} r_n) + b_{n+1} \tag{26}$$

The weight, output variable, and biasing are denoted $w_{n+1}$, $r_n$, and $b_{n+1}$. The computation function is denoted $c$. In the context of neural network learning, the cost function for each labeled specimen $(x, y)$ is determined during the training procedure of training set $\{(x_1, y_1), (x_2, y_2), \cdots, (x_m, y_m)\}$, which comprises m samples, and represented in Equation (27).

$$C(W, b) = \frac{1}{n} \sum_{k=0}^{n-1} \left| h_{W,b}\left(x^k\right) - y^k \right|^2 \tag{27}$$

The hidden layer function is denoted $h_{W,b}$, the input of that layer is denoted $x^k$, and the output is denoted $y^k$. The gradient descent approach can yield favorable convergence outcomes and attain the optimal local value. Consequently, the variables $W$ and $b$ are established, and the updated formula is expressed in Equations (28) and (29).

$$W_{ij}^k = W_{ij}^{k-1} - \propto \frac{d}{d\,W_{ij}^k} C(W, b) \tag{28}$$

$$b_i^k = b_i^{k-1} - \propto \frac{d}{d\,b_i^k} C(W, b) \tag{29}$$

The computation function is denoted $C(W, b)$, the weight is expressed $W_{ij}^k$, and the biasing is expressed $b_i^k$. The scaling factor is denoted $\propto$. The previous layer weight and bias are denoted $W_{ij}^{k-1}$ and $b_{ij}^{k-1}$.

### 3.8.2. An Efficient Clustering Algorithm

Using efficient clustering in LAL facilitates the grouping of comparable logical and activity-based visualizations, thereby simplifying the neural network's acquisition and generalization of patterns from the LAL information. Clustering is a technique that aids in reducing data dimensionality and enhances the efficiency of the learning procedure. Bloom's theory posits that human cognitive processes can be categorized into six levels: memory, understanding, application, analysis, evaluation, and creation. A dual-tiered approach is employed to extract characteristics from data about online learning behaviors. The initial tier pertains to fundamental elements, encompassing login duration, learning duration, frequency of learning, chosen knowledge points, the number of discussions, the number of inquiries posed, the number of questions answered, amount of searches resolved, duration to finalizing the examination, the complete success rate of the study, and homework evaluation, among others. It comprises $n$ elements, denoted as $\{x_1, x_2, \cdots, x_n\}$. The second layer of analysis pertains to high-level features, encompassing metrics such as the extent of completing homework, the precision of homework fulfillment, the learning inquiries, responses to questions, and the solutions to problems. The set $V_{high}$ comprises n el-

ements, denoted as $\{y_1, y_2, \cdots, y_n\}$. The samples can be subdivided into multidimensional characteristics, denoted as x$x_i = \{x_{i1}, x_{i2}, \cdots, x_{in}\}$ and $y_i = \{y_{i1}, y_{i2}, \cdots, y_{in}\}$, where every element signifies a distinct aspect of learning activities, such as the frequency of logging in and the duration of study sessions. Employing a clustering method to partition the pertinent data attributes is possible upon acquiring multidimensional information.

### 3.8.3. Feature Extraction Method

Extracting features from clustered data utilizes a hidden Markov model based on a DNN. This DNN is a type of neural network that contains multiple hidden layers and operates in a forward direction. The input layer of a clustering algorithm is responsible for representing the underlying features of the data. In contrast, the output layer is accountable for defining the typical characteristics resulting from reducing the dimensionality. The activating function utilized for each node is SIGMOD (*sig*), a nonlinear function. Each node's resultant value is nonlinear and expressed in Equations (30) and (31).

$$y_j^s = sig(x_j) = \frac{1}{1 + \exp(-x_j)} \tag{30}$$

$$x_j = b_j + \sum_{i=0}^{n-1} y_j^s w_{ij} \tag{31}$$

The variables are integral components of the neural network architecture. Specifically, $y_j^s$ denotes the nonlinear output of the jth node in the hth layer, while $x_j$ represents the node input value. The computational sigmoid function is denoted $sig(.)$. $b_j$ and $w_{ij}$ correspond to the bias and connection weight between node *j and i*, respectively. The variables for training DNNs are acquired through an iterative training process utilizing the Backpropagation (BP) method for network propagation. The result is shown in Equation (32).

$$J = \{w_i, b_i, w_j, b_j\} = \frac{1}{N} \sum_{i=0}^{N-1} (x_i' - x_i)^2 \tag{32}$$

The fundamental unit employed in this study is the DNN. Each HMM state of the multi-dimensional feature corresponds to a single node of the DNN. The study employed eight-dimensional characteristics as input and incorporated five concealed layers, each comprising one thousand and twenty-four nodes.

Figure 2 illustrates that the DNN input comprises learning activity data. At the same time, the output matches important typical characteristics following dimensionality decrease and cleaning—the neural network's layered processing enhances the differentiating degree of the parts. Certain academic degrees have limited value and are not utilized to their full potential. The hidden layer processes the feature amounts that delineate the learner's attributes. This process guarantees that the last retrieved characteristics effectively reduce dimensionality while preserving the highest level of prejudice. The output value of a node largely governs the activation of each concealed layer node within a deep neural network. The pace of model adaptation is determined by the learning rate. Smaller learning rates result in gradual changes to the weights over time, necessitating more training epochs, whereas with DNN-LALM, greater learning rates result in necessitating fewer quick changes. Expressive capacity and effective model complexity are two ways to classify the complexity of deep learning models. The voices summarize prior research in these fields by analyzing it along four key dimensions: model framework; model size; optimization technique; and data complexity. Among the most crucial hyper-parameters to adjust while fine-tuning a neural network is the learning rate. The difference between a model unable to improve and a model that achieves state-of-the-art performance may come down to the speed at which it learns. The mean of the hidden layer's output is utilized to approximate the feature results to Gaussian dissemination to achieve a Gaussian distribution of features.

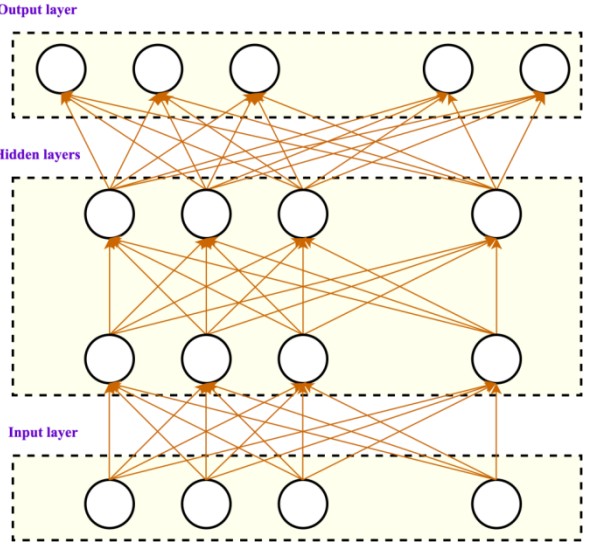

**Figure 2.** DNN architecture for the proposed DNN-LALM.

Each clustering feature acquired is represented as $V_{\propto_i}$, where $\propto_i$ pertains to the i-th training element. The neural network was utilized to obtain the average value of this characteristic in the hidden layers using Equation (33).

$$H_{i,t} = \frac{1}{N} \left( \sum_{k=0}^{N-1} h_{ik} \right) \tag{33}$$

The nonlinear output vectors of the i-th characteristic on the kth layer are denoted as $h_{ik}$. The network characteristic of the parameter is obtained by taking the mean of each concealed layer characteristic shown in Equation (34).

$$F = \frac{1}{N} \sum_{k=0}^{N-1} \sum_{l=0}^{N-1} H_{kl} \tag{34}$$

The hidden layer function is $H_{kl}$, and the total number of samples is $N$. To acquire the efficient feature constituents within the mean characteristic of the concealed layer, the efficient feature following the ultimate dimension reduction is achieved through utilization, shown in Equation (35).

$$E = H_{i,t} - F \tag{35}$$

The hidden layer function is denoted $H_{i,t}$, and the characteristic parameter is denoted $F$. This study examined the impact of advanced DNNs on logical and activity-based learning to improve cognitive abilities. The methodology entailed the utilization of reinforcement learning and unsupervised learning techniques to train deep neural networks on diverse analytical and activity learning tasks. The DNNs were assessed based on their efficacy in enhancing cognitive abilities such as analytical reasoning, judgment, and innovation. The next section presents findings indicating that DNNs have the potential to augment cognitive abilities and offer a promising avenue for the advancement of sophisticated cognitive technologies. The goal of educational DNNs is to improve students' cognitive processes, in effect to help them attain the learning objectives that have been established for each teaching and learning setting. The necessity for a high amount of data and machine resources is one of the primary problems with neural networks and deep learning. Through altering their parameters to reduce a loss function, which assesses the way networks correspond to the data, neural networks learn from knowledge.

## 4. Simulation Analysis and Findings

The EDM Toolbox is an open-source software package offering various data mining algorithms and tools for analyzing educational data [21]. The devices comprise functionalities such as clustering methods, categorization methods, and association rule mining. The platform offers researchers tools for data visualization and preprocessing, aiding in the analysis and interpretation of their data. The EDM Toolbox has been developed to facilitate researchers' quest to enhance educational outcomes by understanding students' learning capabilities.

The Organisation for Economic Co-operation and Development (OECD) conducts the Programme for International Student Assessment (PISA) questionnaires, a comprehensive study evaluating the academic achievements of fifteen-year-old students worldwide [22]. The survey dataset is extensive and covers a broad range of countries. The dataset comprises data about students' academic performance in reading, mathematics, and science, along with contextual and background variables such as educational environment and familial history. The dataset is utilized for simulation analysis to understand the variables that influence academic achievement and provide guidance for educational policy-making. The PISA dataset is a commonly used resource among scholars, decision-makers, and instructors to enhance academic achievements and foster positive student outcomes.

Utilizing a DNN technique has several benefits, one of which is its independence in doing feature engineering. In this method, an algorithm searches the data to identify traits that correlate and then combines their success to encourage quicker learning without being specifically instructed.

The following metrics are analyzed in this simulation analysis:

- Accuracy

The concept of accuracy pertains to the proportion of accurately classified instances within a given model. The computation measures the balance of accurately answered questions in a test or assessment.

- Precision

Precision is a metric that evaluates the accuracy of positive predictions by determining the ratio of true positives to all positive predictions. The computation involves the measurement of the percentage of accurate responses to the overall reactions provided by the student.

- Recall

A recall is the ratio of correctly identified positive instances to the total number of positive cases. The calculation can be derived by determining the percentage of accurately responded items to the overall quantity of items presented in an examination or evaluation.

- F score

The F score is a metric that combines precision and recall in a weighted average, resulting in a more honest evaluation of model performance compared to accuracy as a standalone measure. The computation involves the utilization of the harmonic mean of precision and recall, with higher values indicating superior overall performance.

- Area Under the Curve (AUC)

The AUC is a metric used to evaluate the efficacy of a binary classifier. It is determined by computing the area beneath the Receiver Operating Characteristic (ROC) curve. The metric can be employed to assess the efficacy of a model in forecasting student achievement, with high values denoting superior performance.

Figure 3 showcases the accuracy outcomes (%) of six distinct approaches, namely Support Vector Machine (SVM) [23], Random Forest (RF) [24], Principal Component Analysis (PCA) [25], Naive Bayes (NB) [26], Linear Discriminant Analysis (LDA) [27], and the suggested DNN-LALM, over ten iterations with a range of iterations from one to two hundred fifty. The DNN-LALM method exhibits a consistently higher mean accuracy

than the other methods for the fifty students' performance. The precision of the remaining techniques varies between 73.45% and 91.07% following two hundred fifty iterations. The DNN-LALM method, as proposed, exhibits superior performance compared to alternative methods, with an observed increase in accuracy ranging from 6.5% to 12.47%. The enhanced performance of DNN-LALM can be ascribed to its sophisticated deep neural network structure incorporating logical and activity-based learning mechanisms that augment cognitive abilities, thereby facilitating effective learning and processing of intricate data patterns.

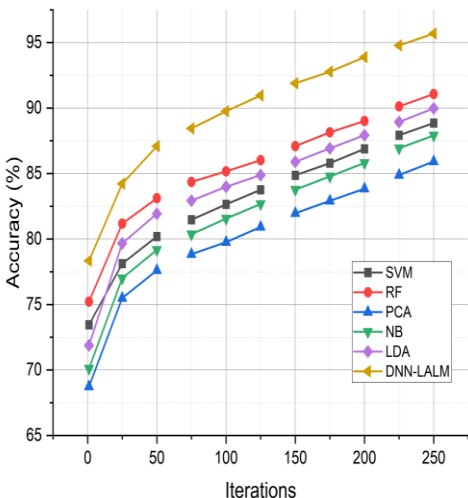

**Figure 3.** Accuracy analysis of the DNN-LALM.

The precision results of six distinct methods, namely SVM, RF, PCA, NB, LDA, and the proposed DNN-LALM, are presented in Figure 4. The data is based on ten iterations with a range of iterations from one to two hundred fifty. The DNN-LALM method exhibits a consistently higher mean precision compared to other methods. The highest precision of 94.6% is attained after two hundred fifty iterations. After two hundred fifty iterations, the precision of the remaining methods varies between 71.39% and 90.93%. The DNN-LALM technique, as proposed, exhibits superior performance compared to alternative methods, with precision improvements ranging from 2.97% to 7.62%. The enhanced performance of DNN-LALM can be assigned to its developed deep neural network architecture that incorporates logical and activity learning to improve cognitive abilities. This architecture is specifically designed to remember and process intricate data patterns efficiently.

The percentage of recall achieved in analyzing student performance through various machine learning techniques is shown in Figure 5. The proposed methodology, DNN-LALM, is evaluated against five alternative approaches: SVM, RF, PCA, NB, and LDA, concerning their precision, recall, and accuracy metrics. In general, it can be observed that the DNN-LALM approach exhibits superior performance compared to the alternative methods across all three evaluation metrics. The DNN-LALM method demonstrates the highest recall scores in nearly all of the iterations. At the two hundred fiftieth iteration, the DNN-LALM model attains a recall score of 95.38%, surpassing the RF method, which is the second-best approach, by 4.18%. This suggests that the proposed methodology exhibits superior proficiency in accurately detecting instances of true positive cases in comparison to the alternative procedures. The proposed DNN-LALM method performs excellently compared to SVM, RF, PCA, NB, and LDA. Specifically, at iteration two hundred fifty, the recall score is improved by 27.12%, 21.3%, 28.6%, 21.53%, and 28.88%, respectively, indicating the effectiveness of the proposed approach. This research investigates the impact of sophisticated deep neural networks on cognitive and behavioral learning, focusing on their computational implications for developing higher-order thinking abilities. The

DNN-LALM approach integrates deep neural networks and logical and activity learning algorithms to enhance the evaluation of student performance.

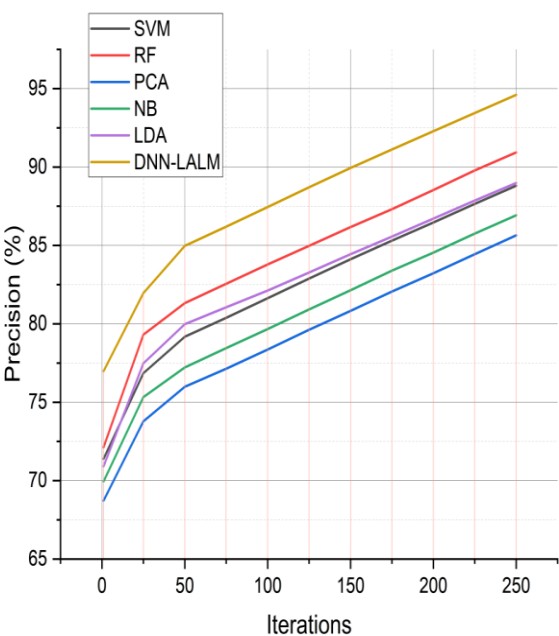

**Figure 4.** Precision analysis of the DNN-LALM.

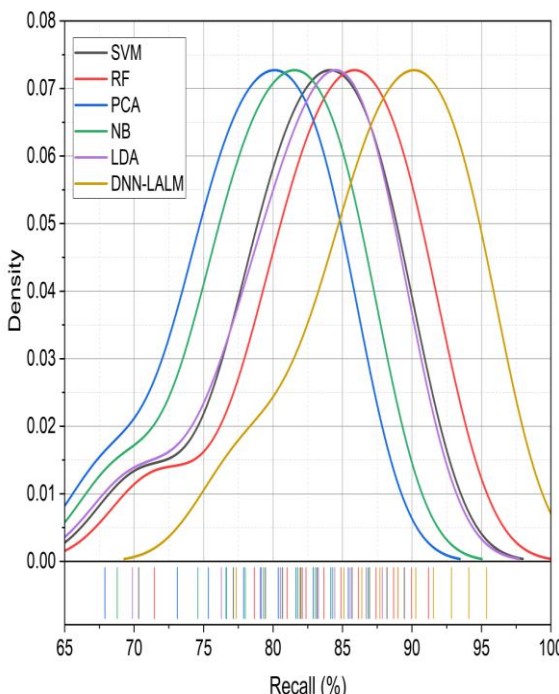

**Figure 5.** Recall analysis of DNN-LALM.

The F score outcomes are expressed in percentage for analyzing student performance through six distinct techniques, namely SVM, RF, PCA, NB, LDA, and DNN-LALM, across various iterations. The F score is a metric utilized to evaluate the precision of a classification model, considering both precision and recall. With an increase in the number of iterations, all of the methods improved accuracy, recall, and F scores. The statistical analysis indicates that the DNN-LALM approach exhibited superior performance compared to other methodologies, as evidenced by the consistently higher precision, recall, and F score values. At

the two hundred fiftieth iteration, the DNN-LALM approach attained a higher F score of 94.42% compared to other methods. This study proposed a methodology for investigating the impact of advanced deep neural networks on logical and activity learning to enhance cognitive abilities. This involved utilizing a DNN-LALM approach integrating deep neural networks with an analytical activity learning model. Figure 6 indicates that the suggested approach yielded a noteworthy enhancement in the evaluation of student performance as compared to conventional methodologies. The aforementioned underscores the potential of employing sophisticated deep neural networks to augment cognitive abilities across diverse domains, such as education.

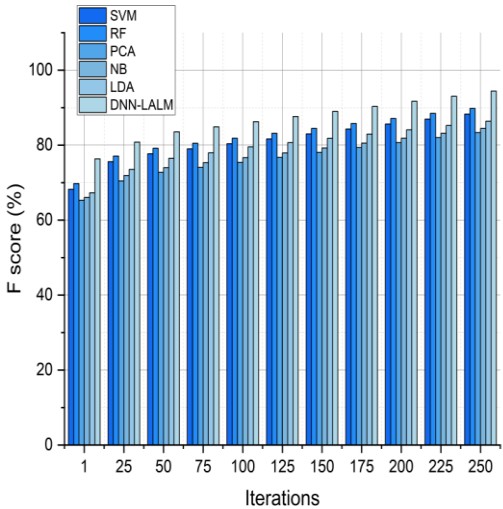

**Figure 6.** F score analysis of the DNN-LALM.

Figure 7 displays the AUC values for six distinct methods utilized in student performance analysis: SVM, RF, PCA, NB, LDA, and DNN-LALM. The area under the receiver operating characteristic curve exhibits a range of values between 0.75 and 0.97, with DNN-LALM demonstrating superior performance compared to the other methods. The DNN-LALM approach shows excellent recall, F score, and AUC metrics compared to alternative methods. The DNN-LALM method performs better than other methods, with improvements ranging from 2.52% to 17.05% for recall, 3.48% to 14.09% for F score, and 5.33% to 21.05% for AUC. The DNN-LALM approach has been developed to investigate the impact of sophisticated deep neural networks on logical and activity-based learning, aiming to improve cognitive abilities. The study's findings indicate that using the DNN-LALM approach can enhance the analysis of student performance more effectively when compared to conventional machine learning techniques such as SVM, RF, PCA, NB, and LDA. The findings suggest that the DNN-LALM model is proficient in capturing the intricate non-linear associations between the input features and output labels, which ultimately results in enhanced performance in the analysis of student performance.

Using a specific dataset, the DNN-LALM algorithm was subjected to simulation and comparative analysis with other classifiers, namely SVM, RF, PCA, NB, and LDA. According to the simulation results, it was observed that DNN-LALM exhibited superior performance compared to the other classifiers regarding accuracy, precision, recall, and F score. The DNN-LALM accuracy mean values for iterations 1, 25, 100, and 250 were 87.16%, 90.03%, 82.39%, and 89.76%, respectively. The results indicate that the average precision, recall, and F score values were 77.43%, 88.23%, and 84.79%, respectively. In general, the DNN-LALM algorithm that has been proposed exhibits potential as a viable method for tasks related to classification.

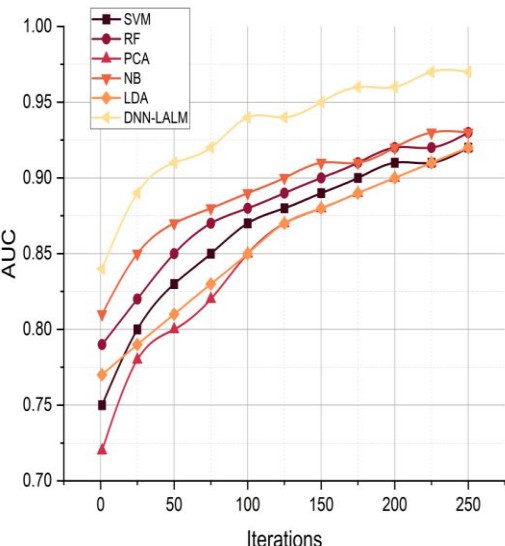

**Figure 7.** AUC analysis of the DNN-LALM.

## 5. Conclusions and Future Study

The significance of fostering students' thinking skills through Logical and Activity Learning is crucial, and its efficacy can be augmented using advanced deep neural networks. The present study introduces a novel approach, DNN-LALM, to evaluate students' academic achievement in logical and activity-based learning. The system employs Deep Neural Networks (DNNs) to forecast students' academic achievement across diverse educational tasks. The simulation findings indicate that DNN-LALM outperformed conventional machine learning techniques, including SVM, RF, PCA, NB, and LDA, regarding the accuracy, precision, recall, and F score percentages. The DNN-LALM technique was superior in performance to the other methods concerning the accuracy, precision, recall, and F score ratios across all iterations. The precision rate attained by DNN-LALM was 78.34% during the first iteration, increasing to 95.69% by the two hundred fiftieth iteration. At iteration one, the SVM, RF, PCA, NB, and LDA methods attained accuracies of 73.45%, 75.22%, 68.73%, 70.11%, and 71.88%, correspondingly. At the two hundred fiftieth iteration, the methods above achieved accuracy rates of 88.86%, 91.07%, 85.92%, 87.91%, and 89.98%, correspondingly. The findings of this research indicate that the DNN-LALM approach holds potential as a method for evaluating learners' academic achievement in the domains of logical reasoning and practical application.

Even so, there exist specific challenges that require attention, including the necessity for extensive datasets and substantial computational capabilities. Future investigations concentrate on enhancing the efficacy of the DNN-LALM approach and tailoring it to various educational pursuits and assignments. The research findings indicate that sophisticated deep neural networks can augment the efficacy of logical and activity-based pedagogical approaches, thereby playing a crucial role in fostering Logical and Activity Learning among students.

**Author Contributions:** Methodology, D.L.; Software, K.D.O. and M.W. All authors have read and agreed to the published version of the manuscript.

**Funding:** This work was supported by Ministry of Education Industry-School Cooperative Education Project, Project Name: Design and Application of Blended Teaching Mode in colleges and universities under the background of digital technology. Project number: 220601590231016.

**Conflicts of Interest:** The authors declare no conflict of interest.

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
