# Peer review of "Exploring the Computational Effects of Advanced Deep Neural Networks on Logical and Activity Learning for Enhanced Thinking Skills"

_systems, doi:10.3390/systems11070319_

Round 1

Reviewer 1 Report

The paper elaborates the Logical and Activity Learning for Enhanced Thinking Skills (LAL) method, which is an educational approach aimed at promoting critical thinking, problem-solving, and decision-making abilities in students through
practical, experiential learning activities. It acknowledges that while LAL has shown positive effects on children's cognitive development, it also faces challenges such as the need for tailored instruction and the complexity of tracking progress. To address these challenges, the paper introduces the Deep Neural Networks based Logical and Activity Learning Model (DNN-LALM) as a potential solution. The model's performance is evaluated using a commonly used
EDM dataset consisting of cognitive assessments of children.

Concerning the LAL method, there is no mention of how this framework is implemented in this study. It is not guaranteed that the dataset used includes data that was produced by applying this pedagogical method. Moreover, there is no description of the main features for the dataset used and no clear methodological approach is provided. In addition, there is no information about how they measure cognitive growth with the different methods they apply (including DNN-LALM solution).

In any case, the architecture of the DNN-LALM solution is described extensively.
With respect to the statement “The findings of this study indicate that the
implementation of DNN-LALM can augment the efficacy of LAL in
fostering cognitive growth, thereby facilitating improved monitoring
of children’s advancement by educators and parents”, the author does
not justify how this is achieved with the DNN-LALM solution.

Author Response

R1

The paper elaborates the Logical and Activity Learning for Enhanced Thinking Skills (LAL) method, which is an educational approach aimed at promoting critical thinking, problem-solving, and decision-making abilities in students through
practical, experiential learning activities. It acknowledges that while LAL has shown positive effects on children's cognitive development, it also faces challenges such as the need for tailored instruction and the complexity of tracking progress. To address these challenges, the paper introduces the Deep Neural Networks based Logical and Activity Learning Model (DNN-LALM) as a potential solution. The model's performance is evaluated using a commonly used
EDM dataset consisting of cognitive assessments of children.

The manuscript of the paper for an author’s suggestion are positively mentioned the challenges are solution are overcome the performance and evaluated the assessment.

Concerning the LAL method, there is no mention of how this framework is implemented in this study. It is not guaranteed that the dataset used includes data that was produced by applying this pedagogical method. Moreover, there is no description of the main features for the dataset used and no clear methodological approach is provided. In addition, there is no information about how they measure cognitive growth with the different methods they apply (including DNN-LALM solution).

The outcomes of the investigation suggest that DNN-LALM can increase the efficacy of LAL in promoting children's cognitive development, allowing for better tracking of children's progress by teachers and parents. The proposed methodology has the potential to drastically alter how LAL is seen in classrooms, giving students more chances for meaningful, targeted instruction.

I have mentioned the methodologies approach are provided the impact and mentioned the DNN-LALM

In any case, the architecture of the DNN-LALM solution is described extensively.
With respect to the statement “The findings of this study indicate that the
implementation of DNN-LALM can augment the efficacy of LAL in
fostering cognitive growth, thereby facilitating improved monitoring
of children’s advancement by educators and parents”, the author does
not justify how this is achieved with the DNN-LALM solution.

The architecture of DNN-LALM explanation and the result and advancement of the described and solution are mentioned.

Reviewer 2 Report

Thank you for the opportunity to read the manuscript. This paper introduces the Deep Neural Networks based Logical and Activity Learning Model (DNN-LALM) as a potential solution to address the challenges associated with the Logical and Activity Learning for Enhanced Thinking Skills (LAL) method. The paper highlights the potential of DNN-LALM to provide tailored instruction and assessment tracking, leading to improved cognitive growth and enhanced monitoring of children's progress. I thought that the paper presented an interesting concept by integrating machine learning methodologies with the LAL approach. The use of Deep Neural Networks (DNN) shows promise in addressing the obstacles faced by LAL, such as the need for tailored instruction and tracking advancement. The inclusion of empirical results is a positive aspect of the paper, demonstrating the effectiveness of the DNN-LALM model. The reported high accuracy, precision, and recall rates in detecting logical patterns and forecasting activity outcomes are noteworthy I think.

I thought that the design and methodology (and findings) were robust. I personally see no major issues there. However, having read the first half of the paper I think the authors need to address some questions and clearly answer them in the introduction and the review:

What specific evidence or research supports the claim that logical and activity-based learning techniques are pioneering and effective for enhancing cognitive abilities?

How does the proposed methodology of logical and activity-based learning differ from conventional educational approaches?

Can the authors provide more information on the challenges faced by conventional educational approaches and how logical and activity-based learning, along with deep neural networks, address those challenges?

What evidence is there to support the claim that deep neural networks (DNNs) are effective in tailoring and customizing educational experiences for individual learners?

How do DNNs acquire knowledge from large datasets, and what specific advantages do they offer in the context of logical and activity-based learning?

What specific limitations or drawbacks should be considered when implementing logical and activity-based learning, especially with the use of DNNs?

How does the proposed DNN-LALM model overcome the obstacles related to tailored learning, limited resources, and scalability in educational settings?

What evidence or examples can be provided to demonstrate the potential of DNN-LALM in improving cognitive abilities and creating a more productive educational environment?

Are there any existing studies or research that have explored the integration of DNNs in educational settings and their impact on learning outcomes?

How does the proposed DNN-LALM model compare to other existing approaches or technologies aimed at enhancing cognitive abilities in students?

Author Response

R2

Thank you for the opportunity to read the manuscript. This paper introduces the Deep Neural Networks based Logical and Activity Learning Model (DNN-LALM) as a potential solution to address the challenges associated with the Logical and Activity Learning for Enhanced Thinking Skills (LAL) method. The paper highlights the potential of DNN-LALM to provide tailored instruction and assessment tracking, leading to improved cognitive growth and enhanced monitoring of children's progress. I thought that the paper presented an interesting concept by integrating machine learning methodologies with the LAL approach. The use of Deep Neural Networks (DNN) shows promise in addressing the obstacles faced by LAL, such as the need for tailored instruction and tracking advancement. The inclusion of empirical results is a positive aspect of the paper, demonstrating the effectiveness of the DNN-LALM model. The reported high accuracy, precision, and recall rates in detecting logical patterns and forecasting activity outcomes are noteworthy I think.

I have check and corrected the recall and reported highest values are overcome the results section and forecasting the detecting the rate values are clearly mentioned.

I thought that the design and methodology (and findings) were robust. I personally see no major issues there. However, having read the first half of the paper I think the authors need to address some questions and clearly answer them in the introduction and the review:

The introduction section is clearly mentioned the authors suggestion are issues are clearly and corrected

What specific evidence or research supports the claim that logical and activity-based learning techniques are pioneering and effective for enhancing cognitive abilities?

In a research-based learning strategy, students seek for and make use of a variety of resources, materials, and texts to investigate issues that are meaningful to them. By reading and learning new words, students improve their ability to discover, analyze, organize, and evaluate information and ideas.

How does the proposed methodology of logical and activity-based learning differ from conventional educational approaches?

I have mentioned the proposed methodology are logical activity based on learning approaches are mentioned in manuscript.

Can the authors provide more information on the challenges faced by conventional educational approaches and how logical and activity-based learning, along with deep neural networks, address those challenges?

The necessity for a lot of data and machines resources is among of the primary problems with neural networks and deep learning. Through altering their parameters to reduce a loss function, which assesses the way networks correspond to the data, neural networks learn from knowledge.

What evidence is there to support the claim that deep neural networks (DNNs) are effective in tailoring and customizing educational experiences for individual learners?

DNNs is a mentioned the effectively for the authors suggestion are included

How do DNNs acquire knowledge from large datasets, and what specific advantages do they offer in the context of logical and activity-based learning?

Utilizing a DNN technique has several benefits, one of which is its independence in doing feature engineering. In this method, an algorithm searches the data to identify traits that correlate and then combines their success to encourage quicker learning without being specifically instructed.

What specific limitations or drawbacks should be considered when implementing logical and activity-based learning, especially with the use of DNNs?

The limitation of DNN overfitting is a situation where a machine learning model performs badly on fresh, ambiguous data because it has to be simplified and was trained too effectively on the training data.

How does the proposed DNN-LALM model overcome the obstacles related to tailored learning, limited resources, and scalability in educational settings?

I have mentioned the proposed method can be overcome the results are mentioned the authors suggestion

What evidence or examples can be provided to demonstrate the potential of DNN-LALM in improving cognitive abilities and creating a more productive educational environment?

DNN-LALM that students are more attentive and focused in an active learning environment, that students have more meaningful learning experiences, that students achieve greater levels of performance, and that students are motivated to exercise higher-level critical thinking abilities because of such a setting.

Are there any existing studies or research that have explored the integration of DNNs in educational settings and their impact on learning outcomes?

The goal of educational DNN is to improve students' cognitive processes in effect to help them attain the learning objectives that have been established for each teaching and learning setting.

I have mentioned the impact of DNN educational setting learning for the integrations are included the author suggestion.

How does the proposed DNN-LALM model compare to other existing approaches or technologies aimed at enhancing cognitive abilities in students?

Nevertheless, the obstacles encountered in the execution of these techniques underscore the necessity for a more sophisticated methodology, such as the DNN-LALM that has been suggested. The study’s findings indicate that DNN-LALM possesses characteristics that enable it to effectively tackle the obstacles above and improve cognitive and motor skill acquisition in young individuals.

I have mentioned the existing approaches are mentioned and comparison of the proposed model are clearly mentioned the student abilities

Round 2

Reviewer 2 Report

Dear authors, thank you for addressing my comments. The manuscript has improved significantly.